# INTERPRETABLE SYNTACTIC REPRESENTATIONS ENABLE HIERARCHICAL WORD VECTORS

## ABSTRACT

The distributed representations currently used are dense and uninterpretable, leading to interpretations that themselves are relative, overcomplete, and hard to interpret. We propose a method that transforms these word vectors into reduced syntactic representations. The resulting representations are compact and interpretable allowing better visualization and comparison of the word vectors and we successively demonstrate that the drawn interpretations are in line with human judgment. The syntactic representations are then used to create hierarchical word vectors using an incremental learning approach similar to the hierarchical aspect of human learning. As these representations are drawn from pre-trained vectors, the generation process and learning approach are computationally efficient. Most importantly, we find out that syntactic representations provide a plausible interpretation of the vectors and subsequent hierarchical vectors outperform the original vectors in benchmark tests.

## 1 INTRODUCTION

Distributed representation of words present words as dense vectors in a continuous vector space. These vectors, generated using unsupervised methods such as *Skipgram* (Mikolov et al., 2013) and *GloVe* (Pennington et al., 2014), have proven to be very useful in downstream NLP tasks such as parsing (Bansal et al., 2014), named entity recognition (Guo et al., 2014) and sentiment analysis (Socher et al., 2013). Although semi-supervised models like *BERT* (Devlin et al., 2018) and *ELMO* (Peters et al., 2018) have demonstrated better performance in downstream tasks, the unsupervised methods are still popular as they are able to derive representations using a raw, unannotated corpora without the need of massive amounts of compute. However, the major problem with nearly all of these methods lie within the word representations themselves as their use can be termed as a *black-box* approach.

These dense representations comprise of coordinates which by themselves have no meaningful interpretation to humans, for instance, we are unable to distinguish what a "high" value along a certain coordinate signifies compared to a "low" value along the same direction. Additionally, this phenomenon also restricts the ability to compare multiple words against each other with respect to individual coordinates. It would appear that the numerical values of a word's representation are meaningful only with respect to representation of other words, under very specific conditions. So, is it possible to create a representation that is not only comprised of coordinates that are meaningful to humans, but is also made of individual coordinates that enable humans to distinguish between words?

Ideally such a representation would be of reduced size and capable of capturing the meaning of the word along absolute coordinates i.e coordinates whose values are meaningful to humans and can be compared against other coordinates. A common absolute that encompasses all words in the vocabulary are the eight parts of speech, namely noun, verb, adjective, adverb, pronoun, preposition, conjunction and interjection. The parts of speech explain the role of a particular word within the given context and with a coarse grained approach, all words can be classified as at least one part of speech. This work, precisely describes the representation of words in a vector space where each coordinate corresponds to one of the eight parts of speech, derived from the pre-trained word vectors via post processing. Such representation would be able to capture the absolute syntactic meaning of

a word. For instance, the syntactic representation for the words "right" and "I" can be visualized in figure 1.

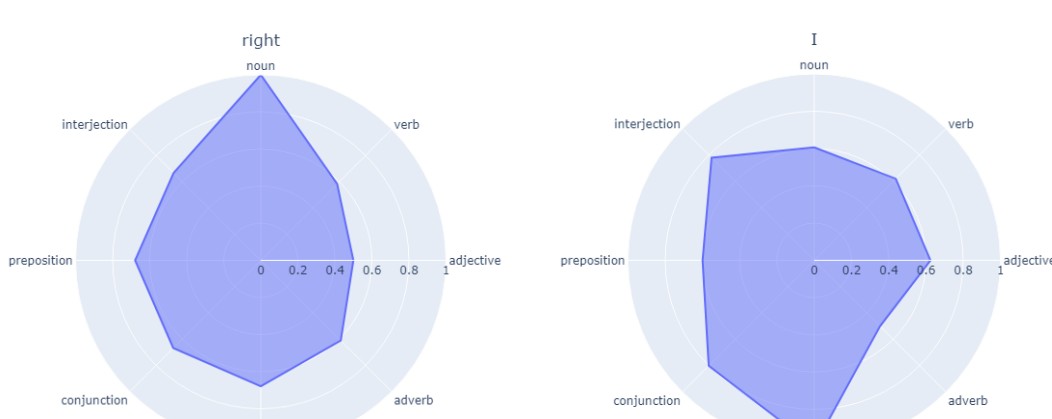

(a) Syntactic Representation for `right`                (b) Syntactic Representation for `I`

Figure 1: Examples of Syntactic Representations

These Syntactic Representations are continuous and highly reduced transformations of the original representations. These representations isolate the syntactic regularities existing within the large original word vectors and when used together in a incremental learning approach as *Hierarchical Vectors*, outperforms the original vectors in benchmark tests. *Hierarchical Vectors* utilize the basics of human learning by emulating the hierarchical aspect of learning. As human learning begins with a simple introduction, branches out to tougher topics and further extends to more complex studies in a hierarchical pattern, the incremental approach used in our work intends to emulate such pattern in word vectors. For this, we have generated unigram Syntactic Representations which act as the foundation of learning and further extend to pre-trained vectors as composite vectors. For any downstream evaluation, the model was trained using a composite representation built using the syntactic representation followed by the original vector.

Our work introduces a novel approach focusing on isolating syntactic regularities within word vectors, deliberately avoiding state-of-the-art contextual information. Consequently, a notable limitation emerges as the model struggles with handling polysemy in words. However, this limitation aligns with the primary aim of this research, laying the groundwork for a fundamental shift in model interpretability and black-box nature of language models. Our methodology, instead of challenging the state-of-the-art directly, aims to use isolated regularities instead of automatically learned hidden representations, thereby questioning the status-quo. This approach aims to bridge the gap between performance and opacity of language models. In the subsequent sections, we elaborate on our methodology and present experimental results illustrating the effectiveness of our approach.

## 2 RELATED WORKS

The previous works of interpretability for word embeddings have generally incorporated the idea of sparse and overcomplete vectors (Faruqui et al., 2015)Subramanian et al. (2018)Panigrahi et al. (2019). Some work have considered the use of semantic differentials to create overcomplete vectors. (Mathew et al., 2020)(Engler et al., 2023) Both of these techniques are based on the idea of transformation into higher dimensional vectors which provide better representation of the pre-trained vector. The fundamental problems with these interpretations are producing transformations which are still inherently uninterpretable i.e they would require further transformations like dimensionality reduction or qualitative analysis to be understood. On, the other end of the spectrum, works such as Berger (2020) and Wang et al. (2023) produce visual interpretations, which are mostly useful for analysis of vectors or the vector space than on downstream tasks. Furthermore, post-processing

works such as Mu et al. (2017) and Aboagye et al. (2022) focus on interpretations for specific tasks rather than on the underlying word representations.

Our work aims to find the balance between these works to introduce two novel ideas. First, a visualized syntactic representation focused on coarse grained eight parts of speech which can be understood by almost anyone. Secondly hierarchical vectors, built as learning from smaller uni-gram Syntactic Representation and followed by larger pre-trained vector, emulating the hierarchical human learning process of starting from simple concepts then branching out to related complexities.

## 3 SYNTACTIC REPRESENTATIONS

The primary objective of generating Syntactic Representations is to convert the dense, continuous word vectors into meaningful vectors with reduced dimensions. These vectors are derived from the original vectors through post-processing methods. Let $V$ be the size of the vocabulary. $\boldsymbol{X} \in \mathbb{R}^{V \times L}$ is the matrix created by stacking $V$ word vectors of size $L$, trained using an arbitrary unsupervised word embedding estimator. This matrix represents the original word vectors that will be transformed into syntactic representations.

The fundamental idea behind constructing Syntactic Representations is to first establish the syntactic subspace and then project the original word vectors onto this syntactic subspace. To build the syntactic subspace, we leverage linguistic knowledge obtained from a pre-constructed linguistic resource, WordNet (Miller, 1995). WordNet is an extensive lexical database of English, categorizing nouns, verbs, adjectives, and adverbs into sets of cognitive synonyms (synsets), each expressing a distinct concept. We gather words belonging to each part of speech from WordNet and supplement them with frequently used words from other parts of speech to construct the syntactic subspace.

Let $\mathbb{N}$ denote the set of word vectors for all nouns in WordNet that are also present in the vocabulary. The Noun coordinate of the change of basis matrix, i.e., the transition matrix $\boldsymbol{C}$ for the syntactic subspace $\boldsymbol{S}$, can then be obtained as:

$$\boldsymbol{C}_{1,:} = \sum_{i=1}^{V} \frac{n_i}{V} \tag{1}$$

where $n \in \mathbb{N}$ represents an individual word vector in $\mathbb{N}$.

These coordinates are calculated across all eight parts of speech and stacked to form the transition matrix for the syntactic subspace, $\boldsymbol{C} \in \mathbb{R}^{(V \cap W) \times 8}$, where $W$ is the vocabulary of all words in Word-Net . Let $v \in V$ be a word in the vocabulary. For $v$ to be projected into the syntactic subspace, it is necessary that $v \in V \cap W$.

$$\boldsymbol{C}^T \cdot \boldsymbol{S}_{v,:} = \boldsymbol{X}_{v,:} \tag{2}$$

$$\boldsymbol{S}_{v,:} = (\boldsymbol{C}^T)^+ \cdot \boldsymbol{X}_{v,:} \tag{3}$$

where $(\boldsymbol{C}^T)^+$ denotes the Moore-Penrose generalized inverse of the transpose of the transition matrix (Ben-Israel & Greville, 2003).

Therefore, the Syntactic Representations for any word $v \in V \cap W$ can be generated as shown in equation 3. We use the equation to generate three models of Syntactic Representations which have been explained in detail in Appendix A

## 4 HIERARCHICAL VECTORS

In this work, we introduce Hierarchical Vectors, composite vectors formed by combining derived Syntactic Representations with the original word representations. The central idea behind Hierarchical Vectors is to emulate the human learning process. As humans, we acquire knowledge and wisdom incrementally, gradually developing our understanding, mastering skills, and adapting to changes in our environment. This incremental learning process, while valuable, inadvertently gives rise to bias as a de facto aspect of human behavior.

However, the predominant approach in most NLP models relies on word vectors that encapsulate syntactic regularities, semantic nuances, and various linguistic subtleties, all compressed into static

representations. While these pre-trained word vectors provide a strong foundation for language understanding, they often lack the adaptability, interpretability, and means for quantifying bias.

To address these limitations, we introduce Hierarchical Vectors, which embrace the concept of incremental learning. This approach entails an initial phase of learning from the reduced Syntactic Representations, followed by the gradual incorporation of the broader original word vectors into the learning process. By employing the incremental learning concept, we aim to isolate the syntactic regularities from the original word vectors, leading to a more refined and enriched representation of the linguistic structures present in the data. This strategic use of incremental learning aids in disentangling the syntactic regularities from the original word vectors, ultimately enhancing their expressiveness and depth. We present the following two models of Hierarchical vectors: Overcomplete and Weighted, which have been explained in detail in Appendix B.

## 5 EXPERIMENTAL SETUP

As our work does not require any raw text corpus for the generation of representations, we use two popular pretrained word vector representations, **Word2Vec** (Mikolov et al., 2013) with 300 dimensions, trained on 3 million words of the Google News dataset and **GloVe** (Pennington et al., 2014) trained on 2.2 Million words of the Common Crawl dataset.

We use the three models of syntactic representations with the two hierarchical vectors to produce six subsequent hierarchical vectors of each of these pretrained vectors. These vectors are then compared against the respective original vectors and their vector siblings. It is important to reiterate that the major intention of this work is to isolate the syntactic regularities from the dense, continuous word vectors to enable interpretation, without improving the performance of the embedding. Rather, we intend to interpret the word vectors without any loss in performance. The notations used henceforth have been explained in Appendix B.3.

## 6 EVALUATION

### 6.1 BENCHMARK DOWNSTREAM TEST

We use the generated Hierarchical Vectors to perform the following benchmark downstream tasks: news classification, noun phrase bracketing, question classification, capturing discriminative attributes, sentiment classification and word similarity. An array of models such as SVC, Random Forest Classifier, Logistic Regression, etc have been used to evaluate the vectors. These tasks are described in detail in Appendix C.

### 6.2 EFFECTS OF TRANSFORMATION

The effects of transformation have been quantified by evaluating their performances to the performance of the respective base vector. Table 1 shows consistent performance of all iterations of Hierarchical Vectors across both base representations. Notably, we observe that the $\mathbf{WO^A}$ vector demonstrates a significant performance boost, excelling in multiple tasks and, on average, outperforming the original Word2Vec vector. Similarly, $\mathbf{GO^L}$ also demonstrates excellent performance, outperforming the Glove vector. We have included the results from the POLAR framework (Mathew et al., 2020) to provide an existing baseline, as it is the most recent work similar to ours. This result includes the highest value between *Word2Vec w/ POLAR* and *GloVe w/ POLAR*.

The exceptions are on the intrinsic word similarity tasks, as presented in Table 7, where the transformed vectors perform worse than the base vectors across various datasets. The assessment of word similarity relies on the Spearman rank correlation coefficient $\rho$, calculated between the human annotated score and the cosine similarity of the word vector pair. However, it's crucial to note that word similarity tests primarily serve as an indicator of vector quality. Additionally, as per the work of Faruqui et al. (2016), which illustrates low correlation between word similarity score and extrinsic evaluations, we can observe the same while comparing Tables 1 and 7. Thus, we contend that these similarity scores can be deprioritized in favor of the vectors demonstrated efficacy in practical applications. The similarity scores are summarized in Appendix D.

Table 1: Performance of the Hierarchical Word Vectors across different downstream tasks. Scores represent the observed accuracy as %.

| Vectors | | Sports Acc. | Relig. Acc. | Comp. Acc. | NP Acc. | Discr. Acc. | TREC Acc. | Senti. Acc. | Average |
|---|---|---|---|---|---|---|---|---|---|
| Word2Vec | | 93.97 | 83.33 | 76.19 | **81.61** | **59.12** | 86.8 | 82.43 | 80.49 |
| Hierarchical Overcomplete Word2Vec | **WO$^A$** | **96.61** | **86.89** | **82.63** | 78.25 | 56.55 | 89.40 | 82.21 | **81.79** |
| | **WO$^I$** | 93.59 | 85.77 | 76.71 | 79.37 | 59.09 | 87.60 | **82.92** | 80.72 |
| | **WO$^L$** | 95.48 | 85.77 | 81.34 | 80.72 | 54.41 | **89.60** | 80.78 | 81.16 |
| Hierarchical Weighted Word2Vec | **WW$^A$** | 94.22 | 84.10 | 75.16 | 79.60 | 59.12 | 86.20 | 82.48 | 80.12 |
| | **WW$^I$** | 94.10 | 85.77 | 75.68 | 79.15 | 59.12 | 87.00 | 82.92 | 80.53 |
| | **WW$^L$** | 94.22 | 84.10 | 74.65 | 79.6 | 59.12 | 86.20 | 82.48 | 80.05 |
| GloVe | | **96.61** | 88.28 | 82.24 | **81.39** | 65.06 | 87.20 | 82.65 | 83.34 |
| Hierarchical Overcomplete GloVe | **GO$^A$** | 95.35 | 87.59 | **84.68** | 77.13 | 60.96 | 85.40 | 82.10 | 81.89 |
| | **GO$^I$** | 95.73 | 88.15 | 81.34 | 79.60 | **65.11** | 88.00 | 82.58 | 82.93 |
| | **GO$^L$** | 95.73 | **88.56** | 84.17 | 80.27 | 60.96 | **90.20** | **83.68** | **83.37** |
| Hierarchical Weighted GloVe | **GW$^A$** | 96.48 | **88.56** | 80.57 | 76.46 | 65.02 | 84.4 | 82.52 | 82.00 |
| | **GW$^I$** | 96.23 | 88.01 | 80.57 | 79.82 | 65.06 | 87.20 | 80.36 | 82.46 |
| | **GW$^L$** | 96.11 | 88.70 | 76.19 | 78.70 | 65.02 | 84.6 | 82.48 | 81.69 |
| POLAR | | 95.10 | 85.20 | 80.20 | 76.10 | 63.80 | 96.40 | 82.10 | 82.7 |

## 6.3 EFFECT OF WORD LIST SIZE

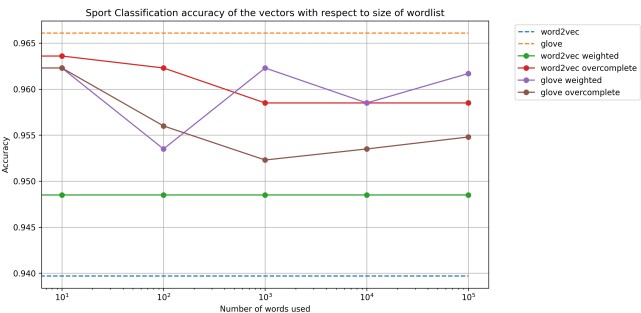

Figure 2: Sport Classification Accuracy per size of word list.

As Syntactic Representations are built using directions recovered form list of words, we have evaluated the effect of word list size on some benchmark tests. As the number of occurring words per each class is different i.e naturally nouns, verbs, adverbs and adjectives occur more frequently than pronouns, conjunctions, prepositions and interjections, we have used repeated instances of words to build the higher list sizes. The results in figure 6 surprisingly showed that the change of accuracy was not consistent with the change in size of word list. We have credited this phenomenon to the larger pre-trained vector having more power over the smaller Syntactic Representation.

The effect of word size list in remaining text classification tasks can be found in Appendix E.

### 6.4 TEST FOR STATISTICAL SIGNIFICANCE

To demonstrate that the observed results were not a case of chance, we have performed a statistical significance test. As suggested by the work of Bouckaert & Frank (2004), we have used 100 runs of random sampling paired with the corrected paired lower-tailed student-t test (Nadeau & Bengio, 1999). We have used this method to check for statistical significance as it considers the ability to replicate results as more important than Type I or Type II errors. Table 2 shows the statistical significance of the generated results.

**Hypothesis:**

$$\mathbf{H}_0 : \mu_{word2vec} - \mu_{hierarchical} = 0$$
$$\mathbf{H}_1 : \mu_{word2vec} - \mu_{hierarchical} < 0$$

(4)

Table 2: Test for Statistical Significance of the results of the benchmark tests. The degree of freedom for the test is **99** and the considered level of significance ($\alpha$) is **10%**. The scores bolded in the table indicate the tests for which the drawn results are statistically significant.

|  | Sports Accuracy | Religion Accuracy | Computer Accuracy | NP Accuracy | TREC Accuracy | Sentiment Accuracy |
|---|---|---|---|---|---|---|
| Word2Vec Mean | 95.17 | 87.63 | 76.53 | 82.37 | 73.56 | 79.74 |
| Word2Vec S.D. | 1.18 | 2.19 | 2.35 | 1.54 | 1.25 | 0.89 |
| Hierarchical Mean | 97.82 | 92.09 | 83.02 | 81.33 | 80.90 | 79.94 |
| Hierarchical S.D. | 0.89 | 1.73 | 2.22 | 1.45 | 1.08 | 1.01 |
| *t*-statistic | -2.14 | -2.14 | -2.34 | 0.69 | -6.07 | -0.21 |
| *p*-value | **0.02** | **0.02** | **0.01** | 0.75 | **0.00** | 0.42 |
| Glove Mean | 95.51 | 87.69 | 78.41 | 81.38 | 72.02 | 79.80 |
| Glove S.D. | 1.18 | 2.20 | 2.41 | 1.65 | 1.31 | 0.96 |
| Hierarchical Mean | 97.84 | 91.23 | 85.73 | 80.43 | 78.05 | 77.82 |
| Hierarchical S.D. | 0.91 | 2.02 | 1.83 | 1.57 | 1.09 | 0.94 |
| *t*-statistic | -2.20 | -1.49 | -2.93 | 0.66 | -4.54 | 1.94 |
| *p*-value | **0.02** | **0.06** | **0.00** | 0.74 | **0.00** | 0.97 |

### 6.5 EFFECT ON SINGULAR VALUES

Following the works of Mu et al. (2017) and Dubossarsky et al. (2020), we perform an evaluation on the singular values of the representation matrices, obtained via Singular Value Decomposition (SVD). The results presented in table 3 display the performance of vectors in a two example tasks accompanied with their largest singular value and condition number. The condition number represents the ratio of the largest singular value to the effective smallest singular value.(Dubossarsky et al., 2020). We can see that the transformation results in high condition number for overcomplete vectors suggests that these vectors may have captured noise, and might need the application of noise reduction techniques.(Ford, 2015) However, the resulting vectors have smaller largest singular number, while maintaining performance, indicating that the post-processing has resulted in successful capture of syntactic regularities and reduction of semantic nuances.

### 6.6 EFFECT ON EMBEDDING SPACE

We observe that the transformed Syntactic Representation provide better syntactic grouping. Even though, the word2vec embedding space is able to group nouns fairly well, the transformed space

Table 3: Performance of the Hierarchical Word Vectors across different downstream tasks. Scores represent the observed accuracy as %.

| Vectors | News Classification Accuracy | Sentiment Accuracy | Largest Singular Value | Condition Number |
|---|---|---|---|---|
| Word2Vec | 84.50 | 82.43 | 777.92 | 5.47 |
| **WO$^\mathbf{A}$** | 88.71 | 82.21 | 198.91 | >100000 |
| **WW$^\mathbf{A}$** | 84.49 | 82.48 | **161.92** | 5.68 |
| Glove | 89.04 | 82.65 | 1689.99 | 4.78 |
| **GO$^\mathbf{A}$** | 89.21 | 82.10 | 1561.38 | >100000 |
| **GW$^\mathbf{A}$** | 88.54 | 82.52 | **681.57** | 10.05 |

better represents all parts of speech, as indicated clearly by the intra-group cohesion and inter-group isolation in figure 3.

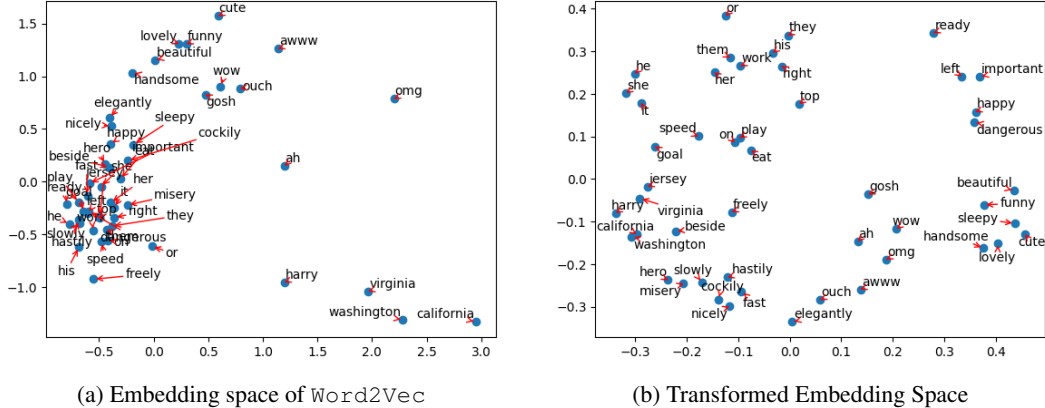

(a) Embedding space of `Word2Vec`    (b) Transformed Embedding Space

Figure 3: Effect of transformation on the embedding space

# 7 INTERPRETABILITY

Our work has stated that the reduced Syntactic Representations allow the words to be compared along an absolute scale while the coordinates also signify the part of speech of the word. To prove this, first we subject the words to a classification experiment based on the words in WordNet. Furthermore, following Faruqui et al. (2015), we conduct a qualitative analysis focusing on individual coordinates.

## 7.1 WORD CLASSIFICATION

The word classification experiment seeks to examine the extent to which the generated Syntactic Representations are coherent to the word's part of speech, as given by an oracle. For this experiment, we consider WordNet to be the oracle. We extract all words present in WordNet to generate respective Interpretable Syntactic Representations, as described in section A.2. The dimension of Interpretable Syntactic Representation with value "1" is selected as the predicted label, for eg, in figure 1 the label of **right** is **noun** and that of **I** is **pronoun**.

As WordNet contains words which have more than one label, we conduct the classification in two ways: Partial and Complete. The Partial classification considers words belonging to only one part of

Table 4: Performance of the Interpretable Syntactic Representations on word classification task. The scores are described as per the four parts of speech in the oracle per each dataset.

| Dataset | Pos | Word2Vec | | | Glove | | |
|---|---|---|---|---|---|---|---|
| | | Precision | Recall | F1-score | Precision | Recall | F1-score |
| Partial | Noun | 0.80 | 0.90 | 0.85 | 0.75 | 0.96 | 0.84 |
| | Verb | 0.82 | 0.80 | 0.81 | 0.88 | 0.80 | 0.84 |
| | Adjective | 0.80 | 0.72 | 0.75 | 0.88 | 0.54 | 0.67 |
| | Adverb | 0.87 | 0.49 | 0.63 | 0.88 | 0.84 | 0.86 |
| **Accuracy** | | **80.33%** | | | 79.21% | | |
| Complete | Noun | 0.76 | 0.82 | 0.79 | 0.72 | 0.88 | 0.79 |
| | Verb | 0.56 | 0.66 | 0.60 | 0.62 | 0.68 | 0.65 |
| | Adjective | 0.74 | 0.67 | 0.70 | 0.81 | 0.52 | 0.63 |
| | Adverb | 0.81 | 0.45 | 0.58 | 0.82 | 0.82 | 0.82 |
| **Accuracy** | | 73.00% | | | **73.58%** | | |

speech while the Complete classification considers all words. The classification results can be seen in table 4. The respective confusion matrices are presented in Appendix F.

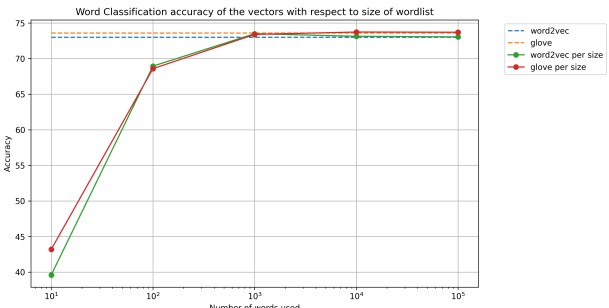

Figure 4: Word Classification Accuracy per size of word list.

As the oracle is restrained to four parts of speech, the interpretation suffers from the same issue. To overcome this, we have presented a coarse grained eight class classification in Appendix G using the Google Text Normalization Challenge - English Language dataset[1]. The 36 tags were conflated to 8 and evaluated using the aforementioned technique. From the results, we can see that the number of pronouns, conjunctions, prepositions and interjections are very low. Thus, as in an even fine grained approach the number of samples per class would be even lower, we have not pursued interpretations in terms of modals, determiners, etc.

The effect of size of Syntactic Representation on Interpretability have been presented in figure 4. We can clearly see that the Interpretable Syntactic Representations drawn from a larger word list preform better than their counterparts.

## 7.2 QUALITATIVE EVALUATION OF INTERPRETABILITY

Following Faruqui et al. (2015), we perform a qualitative evaluation of interpretability. Given a vector with interpretable dimensions, it is necessary that the top-ranking words for that dimension

---

[1]https://kaggle.com/competitions/text-normalization-challenge-english-language

Table 5: Top-ranked words per dimension for Interpretable Syntactic Representations.

| Dimension | Top Ranking Words |
|---|---|
| Noun | vpnvpn, northbrook, florida, michigan, maryland, oregon, nj, illinois, missouri, washington, virginia |
| Verb | personlise, journal, profile, repairs, reorganize, demoralize, marginalize, disassemble, retool, list, decompile |
| Adjective | sporophores, leptonic, savourest, scarious, formless, unspiritual, unipotent, undeformed, unworldly, nonsubstantive |
| Adverb | jacketss, agitatedly, moodily, ponderously, conspiratorially, inelegantly, cockily, somberly, indelicately, bmtron |
| Pronoun | unto, hath, believeth, saith, thy, heareth, knoweth, receiveth, thou, thine, whosoever, doeth, thyself, keepeth, saidst |
| Conjunction | checkour, checkthese, nevertheless, additionally, nonetheless, chymbers, lynters, however, tyrryts, chyrits, also |
| Preposition | across, between, toward, million, around, over, billion, from, through, approximately, onto, towards, ranging, per, into |
| Interjection | ahhh, ahh, hahaha, awww, ohh, oooh, hahahaha, haha, ohhh, omg, hahah, gosh, ooh, awwww, yay |

exhibit semantic or syntactic groupings. As we have worked to isolate syntactic regularities from pre-trained representations, we perform this evaluation to test for syntactic grouping of the top-ranking words along all coordinates.

As shown in table 5, we find that the representations are able to produce excellent syntactic groupings and the produced groups are coherent with the built dimension. Faruqui et al. (2015) was able to present qualitative analysis using semantic groupings without any labels, however our work uses a coarse-grained approach to produce better interpretations based on predetermined labels.

## 8 LIMITATION

The major limitation of our work is **Polysemy**. As we have used non-contextual embeddings as base vectors, our work cannot distinguish senses. However, the counter to this is that our method also does not conflate the word to one part of speech as we use a most likely definition to select the label using Interpretable Syntactic Representation. Furthermore, our work also demonstrates the syntactic boost received by applying isolated syntactic regularities along with the pre-trained vector. This presents a new possible dimension of adding context on top of non-contextual vectors to avoid existing problems in state-of-the-art embedding generators.

## 9 CONCLUSION

In conclusion, our findings offer a novel approach to embedding learning in the realm of natural language processing. By focusing on isolating and refining syntactic regularities within word vectors, we provide a new path forward in the field. Our work demonstrates the potential for presenting compact and visualizable representations which can also be used for downstream tasks. Our work presents the major evidence that word embeddings can be supplemented by isolated regularities to improve performance in specific tasks while maintaining performance overall, potentially suggesting a solution to the status-quo with a new dimension of context on top of non-contextual vectors.

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

## A    TYPES OF SYNTACTIC REPRESENTATIONS

### A.1    ABSOLUTE SYNTACTIC REPRESENTATIONS

These Syntactic Representations are the actual outcomes obtained through post-processing of the original vectors. Unlike conventional normalization, which scales data to a specific range, we intentionally preserve these representations in their unnormalized form. The absolute representations for all parts of speech are constructed based on the words within each category from WordNet, ensuring that these words are also included in the vocabulary of the arbitrary word vector generator.

## A.2 INTERPRETABLE SYNTACTIC REPRESENTATIONS

These Syntactic Representations are thoughtfully presented in an interpretable format, ensuring their accessibility and utility. They are derived using a normalization process that scales them to a range between 0.5 and 1 to ensure that each coordinate in the representation has a value. In this normalized context, the value 1 serves as a clear indicator that the vectors correspond to a specific class.

If $S$ represents the Syntactic Representations, the Interpretable Syntactic Representation, $I$, for coordinate $i$ can be calculated as:

$$I_i = \frac{I_{intermediate} + 1}{2} \tag{5}$$

where,

$$I_{intermediate} = \frac{S_i - min\{S\}}{max\{S_i\} - min\{S\}} \tag{6}$$

## A.3 L2 NORMALIZED SYNTACTIC REPRESENTATIONS

These Syntactic Representations have been meticulously normalized such that their all the square sum of the coordinate values equals to one. The process of normalization is crucial as it ensures that the representations are consistent and comparable.

If $S$ represents the Syntactic Representations, the L2 Normalized Syntactic Representation, $L$, for coordinate $i$ can be calculated as:

$$L_i = \frac{S_i}{\sqrt{\sum_{k=1}^{8} S_k^2}} \tag{7}$$

# B TYPES OF HIERARCHICAL VECTORS

## B.1 OVERCOMPLETE HIERARCHICAL VECTORS

We leverage both the Syntactic Representations and the original vectors of a word to generate Overcomplete Hierarchical Vector for that word. Let $V^s \in \mathbb{R}^{1 \times 8}$ and $V^r \in \mathbb{R}^{1 \times L}$ represent the Syntactic Representation and the original representation of the word $v$, respectively. The Overcomplete Hierarchical Vectors for $v$ can then be computed as follows:

$$V^o = V^s \otimes V^r \tag{8}$$

where $\otimes$ represents the Kronecker product and $V^o \in \mathbb{R}^{1 \times (8 \times L)}$ represents the Overcomplete Hierarchical Vectors.

## B.2 WEIGHTED HIERARCHICAL VECTORS

We use the coordinate values in the Syntactic Representations as weights for the original vector of the word, to generate the Weighted Hierarchical Vector for that word. Let $V^s \in \mathbb{R}^{1 \times 8}$ and $V^r \in \mathbb{R}^{1 \times L}$ represent the Syntactic Representation and the original representation of the word $v$, respectively. Each coordinate of the Weighted Hierarchical Vector for $v$ can then be computed as follows:

$$V_{j:}^w = \sum_{i=1}^{8} \frac{V_{i:}^s \times V_{j:}^r}{8} \tag{9}$$

where $j \in [a, b]$.

The coordinates for the Weighted Hierarchical Vectors are calculated as the element-wise weighted average of the original vector by the Syntactic Representations. The resulting Weight Hierarchical Vector will be of the same dimension as the original vector, i.e $V^w \in \mathbb{R}^{1 \times L}$.

## B.3   Notations

Table 6: Notations and their respective meanings

**NOTATION**

| | | |
|---|---|---|
| **W** | : | Pretrained Word2Vec Representations |
| **WO**$^{\text{A}}$ | : | Hierarchical Overcomplete Absolute Word2Vec |
| **WO**$^{\text{I}}$ | : | Hierarchical Overcomplete Interpretable Word2Vec |
| **WO**$^{\text{L}}$ | : | Hierarchical Overcomplete L2 Word2Vec |
| **WW**$^{\text{A}}$ | : | Hierarchical Weighted Absolute Word2Vec |
| **WW**$^{\text{I}}$ | : | Hierarchical Weighted Interpretable Word2Vec |
| **WW**$^{\text{L}}$ | : | Hierarchical Weighted L2 Word2Vec |
| **G** | : | Pretrained GloVe Representations |
| **GO**$^{\text{A}}$ | : | Hierarchical Overcomplete Absolute GloVe |
| **GO**$^{\text{I}}$ | : | Hierarchical Overcomplete Interpretable GloVe |
| **GO**$^{\text{L}}$ | : | Hierarchical Overcomplete L2 GloVe |
| **GW**$^{\text{A}}$ | : | Hierarchical Weighted Absolute GloVe |
| **GW**$^{\text{I}}$ | : | Hierarchical Weighted Interpretable GloVe |
| **GW**$^{\text{L}}$ | : | Hierarchical Weighted L2 GloVe |

## C   Benchmark Downstream Tests

### C.1   News Classification

Following Faruqui et al. (2015), we use the 20 news-groups dataset[2] and consider three binary classification tasks, namely on the Sports dataset, Religion dataset and the Computer dataset. Each of these tasks consisted of classifying the dataset into two distinct classes: *Hockey* vs *Baseball* for the Sports dataset with a training/validation/test split of 957/240/796, *Atheism* vs *Christian* for the Religion dataset with a training/validation/test split of 863/216/717 and *IBM* vs *Mac* for the Computer dataset with a training/validation/test split of 934/234/777. As in, Subramanian et al. (2018), we use the average of the Hierarchical Vectors of the words in a given sentence as features for classification.

### C.2   Noun Phrase Bracketing

We evaluate the generated Hierarchical Vectors on the NP Bracketing task (Lazaridou et al., 2013), where a noun phrase of three words is categorized as either left beacketed or right bracketed. For instance, given a noun phrase *monthly cash flow*, the task is to decide whether the phrase is {*(monthly cash) (flow)*} or {*(monthly) (cash flow)*}. We used the dataset consisting of 2,227 noun phrases, built using the Penn Treebank (Marcus et al., 1993), in a training/validation/test split of 1602/179/446 data instances. We performed the task by obtaining the feature vector by averaging over the vectors of words in the phrase. SVC (with both Linear and RBF kernel), Random Forest Classifier and Logistic Regression was used for the task, and we use the model with highest validation accuracy for tesing purposes.

---

[2]http://qwone.com/~jason/20Newsgroups/

### C.3 CAPTURING DISCRIMINATIVE ATTRIBUTES

The Capturing Discriminative Attributes task was proposed by Krebs et al. (2018) as a novel task for semantic difference detection, as Task 10 of the SemEval 2018 workshop. The fundamental understanding behind this task was to check if an attribute could be used to discriminate between two concepts. For instance, the model must be able to determine that *straps* is the discriminating attribute between two concepts, *sandals* and *gloves*. This task is a binary classification task on the given SemEval 2018 Task 10 dataset consisting of triplets (in the form *(concept1, concept2, attribute)*) with a train/test/validation split of 17501/2722/2340. As suggested in Krebs et al. (2018), we use the unsupervised distributed vector cosine baseline on the entire data regardless of the training/validation/test split. The main idea is that cosine similarity between the attribute and each concept must enable in discriminating between the concepts.

### C.4 QUESTION CLASSIFICATION (TREC)

To assist in Question Answering, Li & Roth (2002) proposed a dataset to categorize a given question into one of six different classes, eg. whether the question is about a person, some numeric information or a certain location. The TREC dataset is made up of 5452 labeled questions as the training dataset and a test dataset of 500 questions. By isolating 10% as the validation dataset, we have a training/validation/test split of 4906/546/500 questions. As in previous tasks, we construct the feature vector by averaging over the vectors of the constituent words in the question. We train with different classification models and perform the test using the model with the highest reported validation accuracy.

### C.5 SENTIMENT ANALYSIS

The Semantic Analysis task involves testing the semantic property of the word vectors by categorizing a given sentence into a positive or negative class. This test is performed using the Stanford Sentiment Treebank dataset (Socher et al., 2013) which consists of a training/dev/test split of 6920/872/1821 sentences. We generate the feature vector by averaging over all constituent words of the sentence, only on the data instances with non-neutral labels, and report the test accuracy obtained by the model with the highest validation accuracy.

### C.6 WORD SIMILARITY

The Word Similarity test intends to capture the closeness between a pair of related words. We use an array of datasets namely, Simlex-999 (Hill et al., 2015), WS353, WS353-S and WS353-R (Finkelstein et al., 2001), MEN (Bruni et al., 2014) and MT-771 (Halawi et al., 2012). Each pair of data in all of these datasets are annotated a human generated similarity score. We compute the cosine similarity of all of the pairs of words in each of the above dataset and report the Spearman's rank correlation coefficient $\rho$ between the model generated similarity and the human annotated similarity. Only the pairs where both words were present in the vocabulary have been considered for this test.

## D    RESULT OF WORD SIMILARITY EXPERIMENTS

Table 7: Performance of the Hierarchical Word Vectors across different word similarity tasks. Scores represent $\rho$ as %

| Vectors | | Simlex-999 | WS353 | WS353-S | WS353-R | MEN | MT-771 |
|---|---|---|---|---|---|---|---|
| Word2Vec | | **44.20** | **69.41** | **77.71** | **62.19** | **78.2** | **67.13** |
| Hierarchical Overcomplete Word2Vec | $\mathbf{WO^A}$ | 36.89 | 58.8 | 69.58 | 49.79 | 58.46 | 51.39 |
| | $\mathbf{WO^I}$ | 43.94 | 69.13 | 77.39 | 62.00 | 78.03 | 66.99 |
| | $\mathbf{WO^L}$ | 24.00 | 40.67 | 52.64 | 27.60 | 43.02 | 25.88 |
| Hierarchical Weighted Word2Vec | $\mathbf{WO^A}$ | **44.20** | **69.41** | **77.71** | **62.19** | **78.2** | **67.13** |
| | $\mathbf{WO^I}$ | **44.20** | **69.41** | **77.71** | **62.19** | **78.2** | **67.13** |
| | $\mathbf{WO^L}$ | **44.20** | **69.41** | **77.71** | **62.19** | **78.2** | **67.13** |
| GloVe | | **40.83** | 71.24 | 80.15 | **64.43** | 80.49 | **71.53** |
| Hierarchical Overcomplete GloVe | $\mathbf{GO^A}$ | 29.81 | 58.51 | 71.86 | 46.19 | 68.57 | 56.30 |
| | $\mathbf{GO^I}$ | 40.45 | **71.27** | **80.30** | 64.42 | **80.52** | 71.43 |
| | $\mathbf{GO^L}$ | 29.81 | 58.51 | 71.86 | 46.19 | 68.57 | 56.30 |
| Hierarchical Weighted GloVe | $\mathbf{GW^A}$ | 40.33 | 71.24 | 80.15 | **64.43** | 80.49 | 71.44 |
| | $\mathbf{GW^I}$ | **40.83** | 71.24 | 80.15 | **64.43** | 80.49 | **71.53** |
| | $\mathbf{GW^L}$ | 40.33 | 71.24 | 80.15 | **64.43** | 80.49 | 71.44 |

## E    EFFECT OF WORD LIST SIZE

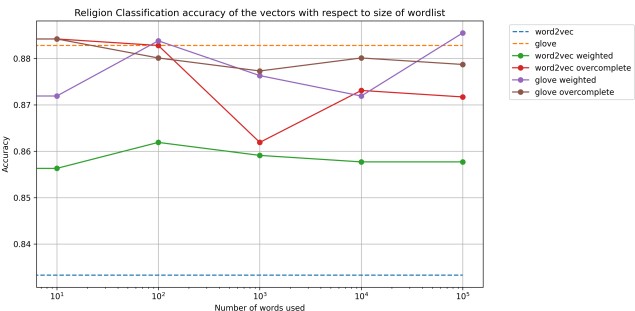

Figure 5: Religion Classification Accuracy per size of word list.

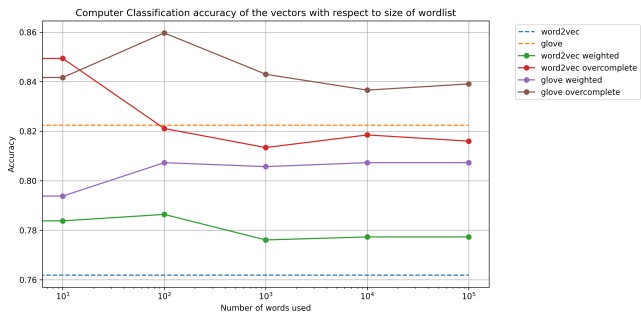

Figure 6: Computer Classification Accuracy per size of word list.

## F  WORD CLASSIFICATION WORDNET

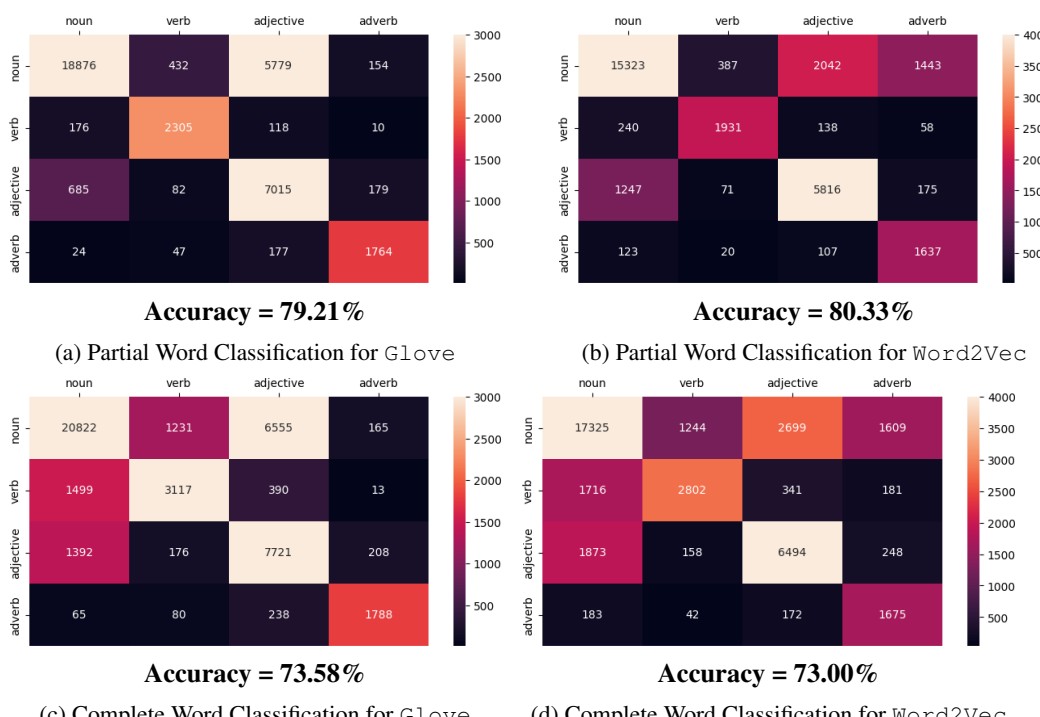

Figure 7: Result of the Word Classification experiment. Accuracy represents the ratio of number of true positive predictions to number of total predictions made in %.

## G    WORD CLASSIFICATION GOOGLE

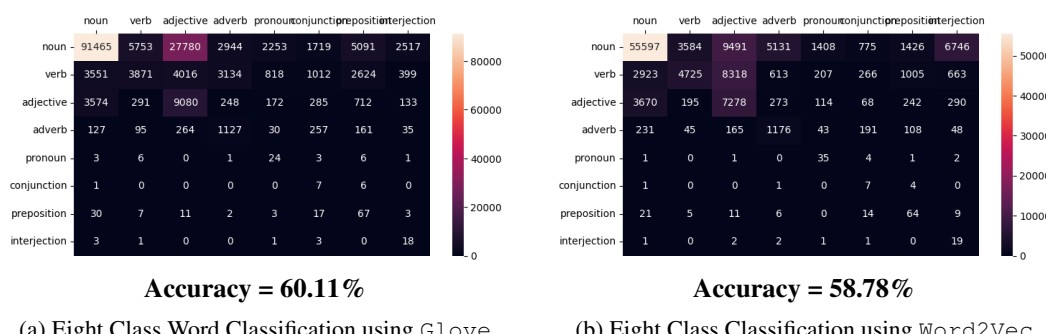

(a) Eight Class Word Classification using `Glove`       (b) Eight Class Classification using `Word2Vec`

Figure 8: Result of the Word Classification experiment. Accuracy represents the ratio of number of true positive predictions to number of total predictions made in %.

