# OpenReview forum: "Syntactic Representations Enable Interpretable Hierarchical Word Vectors"
_ICLR.cc/2024/Conference — Submitted to ICLR 2024_

### Official Review · Reviewer_qs93 · 2023-10-27

**Soundness:** 2 fair
**Presentation:** 3 good
**Contribution:** 1 poor
**Rating:** 3
**Confidence:** 4

**Summary:**

This paper proposes a method for constructing unigram embeddings based on the syntax role of the tokens. They argue the resulting representations to be more interpretable than the representations produced by word2vec or GloVe.

**Strengths:**

In the results, they appear to account for statistical significance in their evaluation, although they don't specify exactly how. The approach is an interesting one from a standpoint of forming hierarchical word embeddings, comparable potentially to approaches like Poincare embeddings, which I encourage the authors to look into. https://arxiv.org/pdf/1705.08039.pdf

They do appear to have results that are competitive with methods like word2vec on classic word similarity metrics and other benchmarks commonly used to test unigram embeddings.

In an era where we often see papers failing to cite a single paper from before 2021, it is actually charming and refreshing to read a paper that doesn't cite anything from after 2018.

**Weaknesses:**

Ultimately, the largest issue with this paper is that it does not address contemporary interests in NLP. It is about a contextless unigram embedding system tested with benchmarks that haven't been used for years.

Even in the era that these citations are from, work like https://aclanthology.org/W16-2506.pdf was questioning the use of the benchmarks used. There was an entire ACL workshop dedicated just to evaluating these types of unigram word embeddings (RepEval). To step backwards into these benchmarks is to disown the work on evaluation and benchmarking done since then.

There is some missing detail about implementation. For example when they mention a normalization process but don't explain how it works. They also don't explain how they determine statistical significance in table 2.

There are two main weaknesses of this work:
1. They failed to justify why anyone should be using contextless unigram embeddings when contextual embeddings work so much better for all applications people are interested in right now.
2. Relatedly, they failed to account for polysemy. This problem is unrecoverable, as far as I can tell, because many words in practice can take on different parts of speech depending on context. For example, "read" could be a noun or verb depending on the context. This is a fundamental flaw in any kind of syntactic encoding system that does not account for context.

I'm also somewhat skeptical of the interpretability results, as the number of classes is so small. There are far more parts of speech than those provided here, which exposes how limited this approach is, as they don't even have things like prepositions or determiners.

**Questions:**

How does the interpretability of these embeddings compare to post-hoc approaches to extracting syntactic information, like probing?

---

> ### Author Response · Authors · 2023-11-23
>
> Thank you for the review.
>
> We do not suggest using unigram embeddings but suggest a need for search of an alternative to the status-quo. The current state has made NLP non-efficient and hard to access. We have proposed a method that can present a new dimension by adding isolated regularities to transform the embeddings as required.
>
> We have added the details to the normalization process, in the Appendix A1 section. We realized the shortcomings with the tests, but did it to check for the quality of vectors, which we have now moved to the appendix section.
>
> Regarding polysemy, that is a major limitation of our work. Although, our work cannot distinguish senses, it does not conflate any wore to one sense. It raises the potential of being able to add context as an isolated regularity on top of contextless embeddings.
>
> Regarding the interpretability issue, we were limited to 4 classes as the oracle, WordNet is also limited to the same classes. However we have presented a newer, more detailed classification report and a full eight class classification in Appendix Word Classification Google, where we show the low number of words in other classes impact the entire classification. Also, as we wanted to present a better visualization of Interpretable embeddings, we went forward with eight parts of speeches as it is easily understandable.
>
> Our work is more interpretable in a visual sense but lacks to the accuracy of probing. We provide a sort of "looking through the models eye" in terms of interpretability, so eight parts of speech was the considered better path forward.
>
> The new version of our work includes addition of statistical significance test, new baseline model, tests per word size, singularity value tests and much more. We have also updated the title to suggest that syntactic representations are the interpretable ones.
>
> We would love to hear more from you.
> Regards.

---

### Official Review · Reviewer_CNTm · 2023-10-31

**Soundness:** 3 good
**Presentation:** 3 good
**Contribution:** 2 fair
**Rating:** 5
**Confidence:** 4

**Summary:**

This paper introduces a new postprocessing method for embedding learning that transforms word vectors into syntactic Representations where each coordinate corresponds to one of the eight parts of speech. The resulting representations are interpretable with each new coordinate having a distinct meaning with respect to the newly defined basis. The authors further introduce hierarchical word vectors derived from these syntactic representations. Experiments on a wide variety of tasks generally show improvements.

**Strengths:**

1. The authors clearly described the background and motivations needed to understand the proposed postprocessing technique.
2. The enduring challenge of interpretability in distributed representation, where meaning is entangled across all coordinates, is addressed in this paper. The authors introduced a new mechanism to convert word vector embeddings into interpretable representations by defining a new basis that is spanned by the eight parts of speech vectors.
3. The authors performed both intrinsic and extrinsic tasks to show that the transformed word embeddings keep their meaning and improve performance on downstream tasks.

**Weaknesses:**

1. No uncertainty/confidence/error bars on experimental results, or significance testing.
2. The experimental results were compared against a simple baseline thus the original embedding. It never showed how it compared against existing baselines.

**Questions:**

1. How sensitive is your proposed method to the size of the word list used to compute the eight parts of speech directions? Providing a similar plot shown in Appendix I of https://openreview.net/pdf?id=TkQ1sxd9P4 should be enough. You could check its sensitivity on an intrinsic or extrinsic task.
2. Glove and Word2vec have been shown to have some inherent structural profile with most of the words being clustered along the long principal component (https://arxiv.org/pdf/1702.01417.pdf). After applying the proposed postprocessing technique could you measure how the structure of the space changes by providing a before and after number of the largest singular value?
3. One experiment to further show how useful the transformed space of your method encodes semantic and syntactic information would be to perform a cross-lingual alignment task between two monolingual embedding spaces. A simple way would be to measure the before and after condition number and singular value gap between the two spaces and report it. Check this paper https://aclanthology.org/2020.emnlp-main.186.pdf on condition number and singular value gap between two language spaces.
4. Does your proposed method enforce an orthogonality between the new basis vectors?
5. Could you include a visualization plot of the before and after postprocessing of the word embeddings?

---

> ### Author Response · Authors · 2023-11-23
>
> Thank you for the review. We have taken the suggested weaknesses and improved our work by adding statistical significance test, new baseline model, tests per word size, singularity value tests and much more. We have changed the title to signify that Syntactic Representations are the ones that are interpretable.
>
> As suggested, we have performed the size test to test sensitivity to size, where we have observed that the performance of Syntactic Representation is dependent on size but that of hierarchical vector is not. We have attributed this to the pre-trained vectors heavier influence on the hierarchical vector. The syntactic representations were sensitive to size to an extent but after the point of 1000 words, due to having to impute many words as WordNet only has four classes, the sensitivity fell.
>
> We have performed the suggested structural test by providing the value of largest singularity and have observed that transformation decreases the value suggesting a potential decrease in noise and semantic nuance. We have presented this in the updated paper. This experiment has shown that we can increase a specific potential of the embedding while keeping the overall performance consistent. As a new dimension, we believe that this method could possibly be replicated to add context to contextless embeddings.
>
> No, we have not ensured orthogonality in the bases.
>
> We have added a plot of the before and after the transformation. The plot clearly shows that the compact grouping before transformation is converted to cohesive intra-class grouping and isolated inter-class grouping.
>
> We would love to hear more from you.
>
> Regards.

---

### Official Review · Reviewer_AnLo · 2023-11-02

**Soundness:** 3 good
**Presentation:** 3 good
**Contribution:** 3 good
**Rating:** 5
**Confidence:** 3

**Summary:**

The word2vec and Glove word embeddings are post-processed in a way that words with identical POS tags will occur in the same subspaces of a vector space. These vectors are tested in a variety of NLP tasks, from similarity to sentiment analysis to question answering and produce good results.

**Strengths:**

Attempting to understand word embeddings by imposing linguistic structure on them. Testing the results in a large range of tasks. Obtaining better results.

**Weaknesses:**

There is a last part to the paper where interpretability is discussed and measured. I did not understand this part and their measure of interpretability. In particular, how do you do the following?

"For assessing the interpretability of our model, we select words from WordNet and subject them to
evaluation using the Interpretable Hierarchical Syntactic Representations."

**Questions:**

Can you please explain why the improved vectors do better in the tasks? What is the intuition behind it? Why should  a noun similarity taks be improved if the noun vectors are grouped together in one part of the space?

---

> ### Author Response · Authors · 2023-11-23
>
> Thank you for the review. We understand about your concern about the last part on interpretability. We have updated the interpretability part to now include an updated version of the classification test. In this test, we extract the words from wordnet and collect their actual pos tag as per WordNet. Then we generate the Interpretable Syntactic Representation for the word, the dimension with value 1 is most likely to be the pos of the word, so it is selected as the predicted pos tag. Using the actual and predicted tags, we calculate the precision, recall, F1-score and eventually total accuracy in the updated work.
>
> The intuition behind the improved vectors doing better in tasks is the hierarchical aspect of human learning. As humans, we first identify a topic, know the basics and eventually delve into the related underlying complexities. However, in NLP we have used models which automatically learn the entire hidden representation. Although this method has been useful in many tasks, it has presented newer issues like those of interpretation and biasness. The main idea of hierarchical vectors doing better is that the unigram syntactic representations help to reduce the unneeded semantic nuances and noise, which in turn increases syntactic abilities while keeping the performance consistent overall.
>
> With respect to the final question, we expected the syntactically similar words to be grouped together, rather than perform well on similarity tasks, as shown in the Qualitative Evaluation section. We have demonstrated through plots that our work was able to provide better groupings but the similarity tests did not show an increase in performance. We did have an improvement in the extrinsic tests, which we have attributed to the aforementioned intuition of hierarchical learning.
>
> We have updated the evaluations to include tests per word size, singularity value tests and much more. We have changed the title to signify that Syntactic Representations are the ones that are interpretable.
>
> We would love to hear more from you,
> Regards.

---

### Meta-Review · Area_Chair_HDLM · 2023-12-07

**Metareview:**

This paper presents a method to transform an existing pre-trained word embeddings into syntactic embeddings that make the embeddings better in quality and interpretability. The method proposed is a projection based method based on existing lexical resources like Wordnet. The main issue with this work is that it is really behind in the time to where the community is right now. The utility of non-contextual word embeddings is a big question, and the evaluation benchmarks being used are sets that are quite obsolete now. When contextual embeddings or just pre-trained models provide very strong results on downstream tasks, it is questionable to invest work in non-contextual embeddings that clearly have knows limitations like polysemy. Regarding intepretability, the clustering of similar words is not enough to really prove that vectors are more interpretable. The plots generated differ heavily based on the set of words selected to generate the plot.

**Justification For Why Not Higher Score:**

This paper has severe issues with experiments and baselines.

**Justification For Why Not Lower Score:**

n/a

---

### Decision · Program_Chairs · 2024-01-16

Reject